# Web Scraping Scientific Repositories for Augmented Relevant Literature Search Using CRISP-DM

**Hossam El-Din Hassanien** 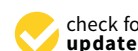

Department of Computer Science, Electrical and Space Engineering, Luleå University of Technology,
SE–971 87 Luleå, Sweden; hossam.hassanien@ltu.se

**Abstract:** Scientific web repositories are central cyber locations where academic papers are stored and maintained. With the nature of the unstructured and semi-structured information/metadata within these repositories, literature analysis for scholar writing becomes a challenge. Correspondingly, applying CRISP-DM poses a stance to address this challenge through formulating a rather augmented process for a relevant literature search. However, almost all repositories do not have a straight forward method where metadata could be extracted for preliminary data processing being applied as part of the CRISP-DM process. Additionally, most repositories do not follow open access standards. Until the time this paper was published, the topic of the augmented, relevant literature search had seen a methodological progress only, with the inability to apply the underlying methods on a larger scale, given data access constraints to open access repositories. The aim of this paper is to propose CRISP-DM as an augmented research methodology with a focus on web scraping as part of the data processing step. To substantiate the proposed methodology, a play role case study is conducted. This then works on alleviating these restrictions, as well as encouraging the wider adoption of the augmented analysis process for a relevant literature search within the research community.

**Keywords:** web scraping; web crawling; CRISP-DM; text mining; relevant literature search; research methodology

---

## 1. Introduction

Typical literature review processes are the ones that work on uncovering what is already known in the body of knowledge, setting out academic debates towards state-of-the-art advancements. These processes are of utmost importance towards initiating any research studies. These processes had existed for decades to assist researchers through the guidelines of academic learnings [1–3]. Despite the benefits gained from effective methodological processes, the execution can still be classified as daunting. Academic writers, especially when conducting literature reviews, suffer from several problems. The following are amongst the top problems typically faced [4]:

- Lack of consistency in the reported results
- Potential towards uncovering flaws within previous research (based on design, data collection instruments, sampling, interpretation, etc.)
- Research may have been conducted on different data populations, which could lead to uncertainty about interpretation of previous studies' findings
- Etc.

For that matter, this leaves several open questions in the minds of academic writers, which then expose some of the main traits of academic publications, hence hindering quality. These traits are: breadth and depth, rigor and consistency, clarity and brevity, furnishing the base for effective analysis

and synthesis [5]. These traits are ones that lay foundations towards answering how one piece of research builds upon another. For these reasons the exercise deems to be in many cases overwhelming.

By examining similar literature review processes; it could be observed that these processes deliberately map to the taxonomy of educational objectives ("a.k.a Bloom's Taxonomy") for obvious reasons of enforcing educational knowledge. Hence, this helps in setting the compass towards more elaborate educational research objectives being derived by combining the processes traits and the taxonomy objectives (highlighted in **bold**):

- Breadth and Depth of **Knowledge**
- Rigor and Consistency of **Comprehension & Application**
- Clarity and Brevity of **Analysis, Synthesis & Evaluation**

This paper is then structured as follows: the Section 2 discusses research motivation behind this scholar work, as well as the resultant questions to be investigated. Section 3 correspondingly explains the adopted research methodology paving the road towards realizing the sought-after outcome of augmenting a relevant literature search. Section 4 highlights the methods employed towards addressing the research questions being discussed, where a proposed web scraping tool of choice could be easily used to extract information. This section also stages an advanced analytics tool that works on elevating the levels of productivity of researchers to increase the adoption of the augmented relevant literature search methodology. Section 5 investigates the application of the proposed methods through a play approach for which a researcher would be inquiring a basket of journals for a specific topic. This section also illustrates the extracted results from the applied methods. For this, the corresponding conclusions are to be highlighted in Section 6.

## 2. Research Motivation & Questions

Visual exploration of bibliographic databases postures a key role towards maintaining the transparency, quality and consistency of the systematic reviews through specific software (CiteSpace, VOSviewer, etc.) [6]. This had contributed significantly towards reinforcing advancements which could be mostly visible in biomedical journals by applying systematic review methodologies [7]. Despite the fact, these principles could also be very relevant to almost any other knowledge domain embarking on achieving the earlier explained educational research objectives.

On the other hand, advanced analytical algorithms also have high aptitude towards complementing these methods of visual exploration. These signs could be observed through the propositions of applying the combinations of text mining techniques to reduce the time reviewing relevant literature [8]. This implicitly addressed a methodological approach that works on identifying relevant literature searches through extractive summarization.

However, the spoken-of advanced analytical methods, as well as the visual exploration software, had somewhat suffered from the user interface lack of flexibility, as well as the restriction to data access. For instance, these advanced analytical methods had been set to retrieve data from open access repositories only; with an example of downloading a research paper corpus from a corpus database like Semantic Scholar (http://labs.semanticscholar.org/corpus/). This corpus database exposes both data samples, or the full data set through API call or Command Line Interface that is restricted to open access repositories (please see Appendix A.1). Likely, visual exploration software sees that the primary source of input data are only repositories like the Web of Science and PubMed [9].

These methods are without a doubt very valuable to researchers. Still, this lack of flexible user interface and restriction to data access poses key obstructions to executing systematic reviews in a timely manner. The sought-after stage of augmenting systematic reviews deems to be a challenge as a result.

Correspondingly, these earlier spoken-of pointers spark the discussion for the following interrogation: How can researchers extend the reach of applying extractive summarization for a relevant literature search beyond open access repositories, if not all scientific repositories especially,

that many repositories do not follow open access standards? In addition, whether the technology landscape possess rather user-friendly interfaces enabling researchers to execute wider advanced analytical methods serving these principles of systematic reviews?

## 3. Research Methodology

In order to reinforce the above-discussed educational research objectives; the Cross-Industry Standard Process model for Data Mining (CRISP-DM) is being adopted as the research methodology for executing principles of systematic reviews [10]. Data mining is the knowledge discovery process which analyses large volumes of data from various aspects and summarizes it into useful information [11]. The data mining process is digested to formulate the CRSIP-DM process model. This model depicted in Figure 1 is composed out of six main iterative steps (Business Understanding, Data Understanding, Data Preparation, Modeling, Evaluation and Deployment) [12].

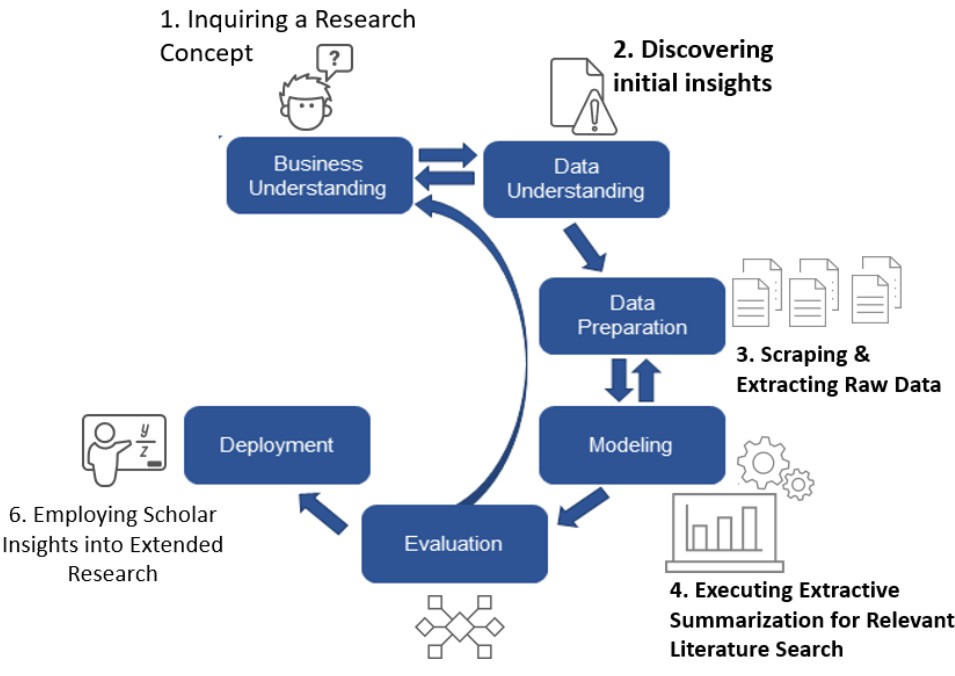

**Figure 1.** Phases of the Cross-Industry Standard Process model for Data Mining (CRISP-DM), being a Process Model for Augmented Systematic Review and Relevant Literature Search.

The ability to execute and follow this process model as a research methodology simplifies one of the most important steps in any literature review process, i.e.: the literature search. The step of "business understanding" sets the compass towards a particular research topic, for which academic writers would be able to address a number of Heilmeir's catechism questions [13]. Following the steps of "data understanding" and "data preparation" surfaces to be the focus of this paper through the web-scraping of data. On the other hand, investigating user friendly interfaces undertaking the step of "modeling" comes as an applied extension to the focus of this paper. Hence, this leads to the end product of augmenting an extractive summarization of a relevant search by "evaluating" the analyzed information that is to be later "deployed" into further research activities.

This then requires an elevated level of flexibility in applying the methodology, for instance when specific sets of publication outlets are being reviewed (reviewing 10 top most journals versus a specific topic). The purpose of this publication is to focus on applying more simplistic methods to introduce this level of elevated flexibility. Hence, encouraging mass adoption of the extractive summarization of the relevant literature search methodology at hand. This paper by turn tackles the following stages of the CIRSP-DM process: (1). Data Understanding, (2). Data Preparation and (3). Modeling by

conducting a case study where specific keywords are to be used for querying a basket of Information Systems journals. Henceforth, this adds a level of augmentation to the evaluation step leading towards efficient scholar writing.

## 4. Research Methods

This section explains the methods used to address the two main questions of how researchers could extend the reach of applying an extractive summarization for a relevant literature search beyond open access repositories, as well as an example of a user-friendly solution that could be used to apply advanced analytics algorithms serving the purpose of systematic reviews.

As explained by Wirth et al. [10], the step of Data Understanding that starts with "*an initial data collection and proceeds discovering first insights into the data, or to detect interesting subsets to form hypotheses for hidden information*". Data Preparation on the other hand "*covers all activities to construct the final dataset (data that will be fed into the modeling tool(s)) from the initial raw data*". That being said, the first subsection illustrates how user-friendly web-scraping interfaces could be used to address the step of discovering insights into the data, as well as to construct the dataset to be used in modeling; where relevant literature would be uncovered. As an example, metadata gets to be extracted from Google Scholar as a test case. Hence, posing the stance towards the ability to extract metadata from any scientific web repository.

Once this initial raw data is furnished, the modeling step (covered in the second subsection) begins to execute the process of enabling the extractive summarization of the process relevant literature, with an example of applying the similarity algorithm.

### 4.1. Web Scraping (Data Understanding & Preparation)

This section focuses on simplifying the stage of Data Preparation for an automated academic literature content extraction [14–16]. Earlier literature had explained how readily available web content could be re-cycled as part of various research processes at different stages as well as the top corresponding challenges. However, The process of automated content extraction (widely known as "Web Scraping", a.k.a "Web Crawling") takes place over three broad steps: (1). Site Access, (2). HTML parsing and contents extraction and (3). Output building [17].

The current state-of-the-art web scraping solutions and technologies offer a wide variety of simple easy to use tools. These solutions came a very long way from a past that was crowded by complexities impeding content extraction; especially for average skilled users. Many of these methods relied on API and/or code-based methods to extract content from designated web sites. The traditional complexity problems in dealing with these APIs and code-based structures could be explained in the convoluted technicalities behind the three steps within the process. For a given user to scrape or crawl content he/she would usually have to hard code many parameters starting with the web-site URL to be crawled ("Site Access"). The problem here lies in the lack of dynamic assignments of URL parameters given the fact that web-site URLs are usually hard coded. This very problem blows out of proportion, especially in the following stage of "HTML parsing and content extraction". HTML parsing and content extraction usually takes place by analyzing the anatomy of web-sites and the underlying content placeholders. A traditional example is where a user would have to understand the xPath structure of the web-site (please see Appendix A.2), as well as know its name in order to include/exclude it for content extraction [18]. Nowadays many of the browsers offer built-in development tools that are designed to help web developers to browse through the structure of a given web page. The value of the development tools is definitely very high for web developers. However, when it comes back to the case of extracting web content from scientific repositories the process remains complex due to the hierarchical xPath structures depicted in Figure 2a,b.

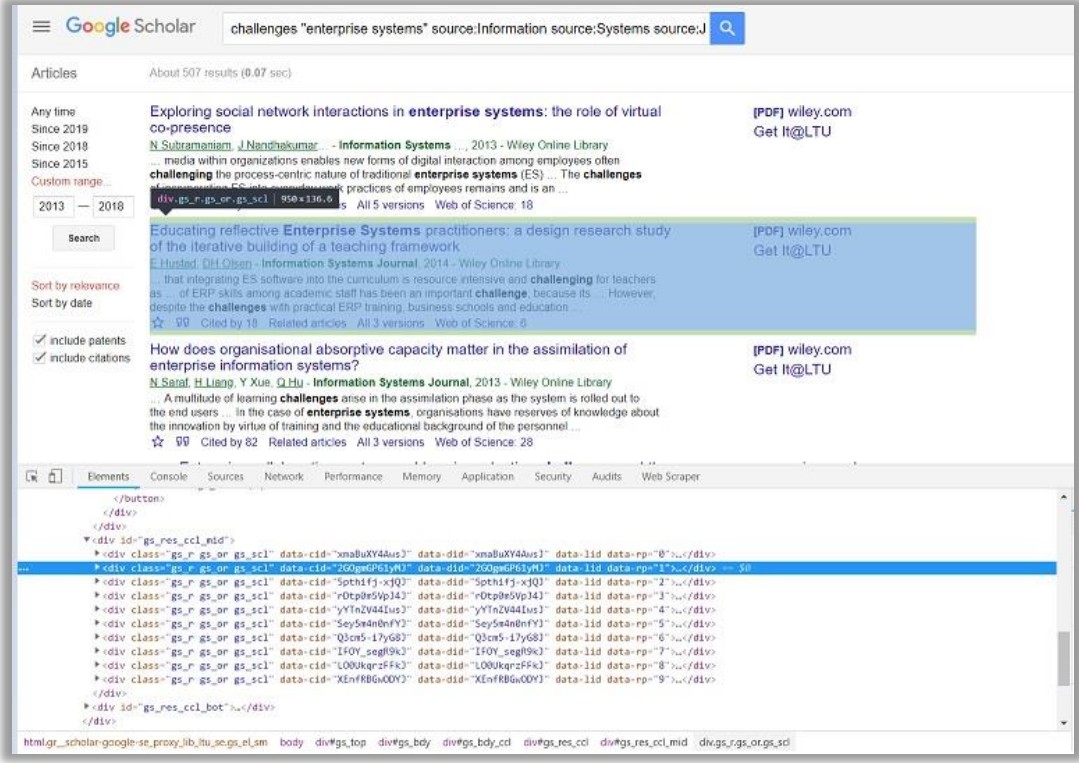

**Figure 2.** Google Chrome's Inspect Element panel dissecting a webpage anatomy: (**a**) parent path highlighting a particular literature search hit; (**b**) highlighting a sub-component of the parent search hit.

Over time, many of the web scraping tools identified in literature [15] had been simplified in order not to only provide free utilities, but also to code free solutions. In our case an extension to the google chrome browser (known as Web Scraper– please see Appendix A.3) had been used to overcome the complexities with HTML parsing and content extraction. The tool provides a set of utilities that enable given users to point-and-extract content of interest by a providing pagination facility to scrape multiple pages as depicted in Figure 3a, visualize the scraping logic through what is known as a selector graph, etc. as depicted in Figure 3b. The output building in this case takes place by exporting the extracted content into csv files which would later be rechanneled into the data preprocessing and modeling.

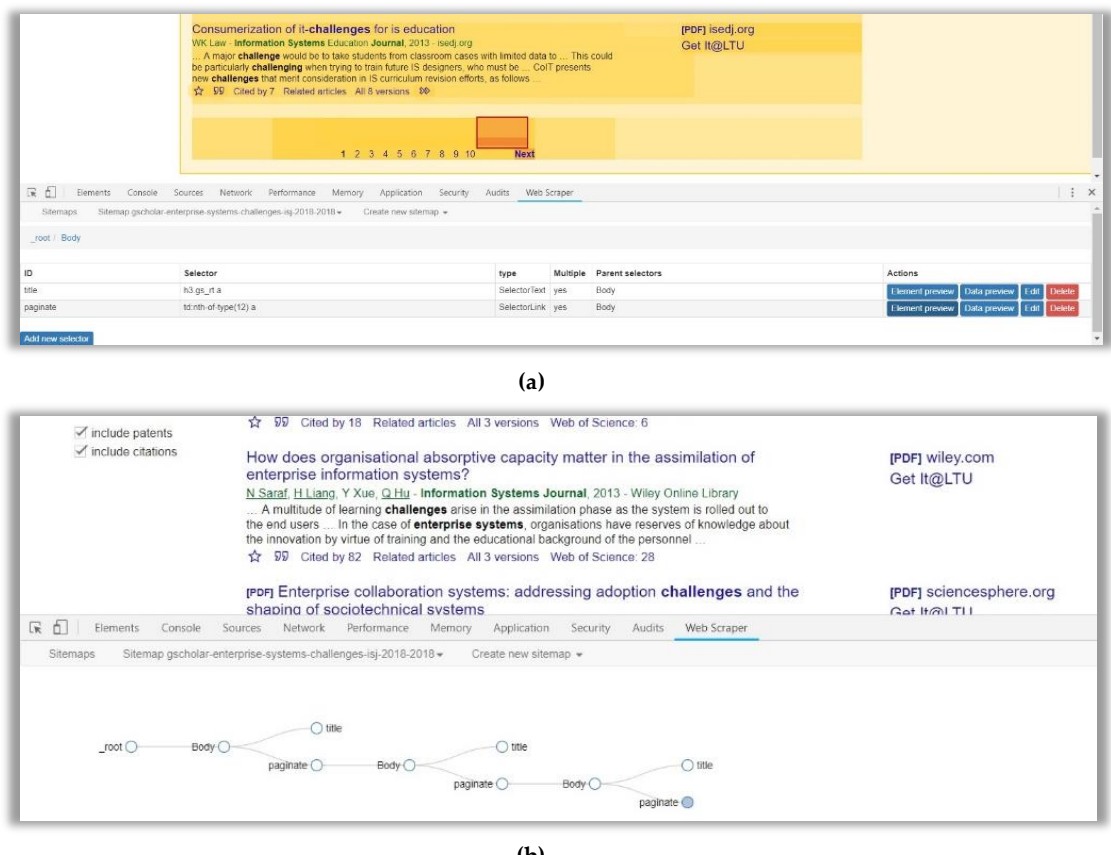

(a)

(b)

**Figure 3.** (**a**) Automated pagination; (**b**) scraping graph visualizing the content extraction logic.

### 4.2. Mining & Analyzing the Relevance of Literature (Modeling)

Conducting the stage of modeling requires a simple yet robust machine-learning platform that would enable users to efficiently manage their productivity. That is of high importance, enabling researchers to augment the process of the literature search, with an aim towards focusing on more important objectives derived from the publications traits and learning taxonomy illustrated above.

The machine learning arena is full of tools that address the different levels of user maturity and understanding of the domain [19]. For the processes of mining and analysis of data for the relevant literature search, a moderately complex tool would be required to aid the ability to apply different text mining tasks. In our case, Rapidminer had been the tool chosen for its good and easy to use graphical capabilities, as well as its out of the box text mining features. If the user is a data mining beginner, KNIME could also be used. As a continuum to the web scraping exercise, the below illustrates a high-level view towards applying text mining.

The above could then be scaled to easily replicate the relevant literature search and summarization methodology proposed in literature while encouraging mass adoption through ease of use. To illustrate

the same, Figure 4a,b shows a high-level text mining process that is built to identify term similarity between the documents being scraped.

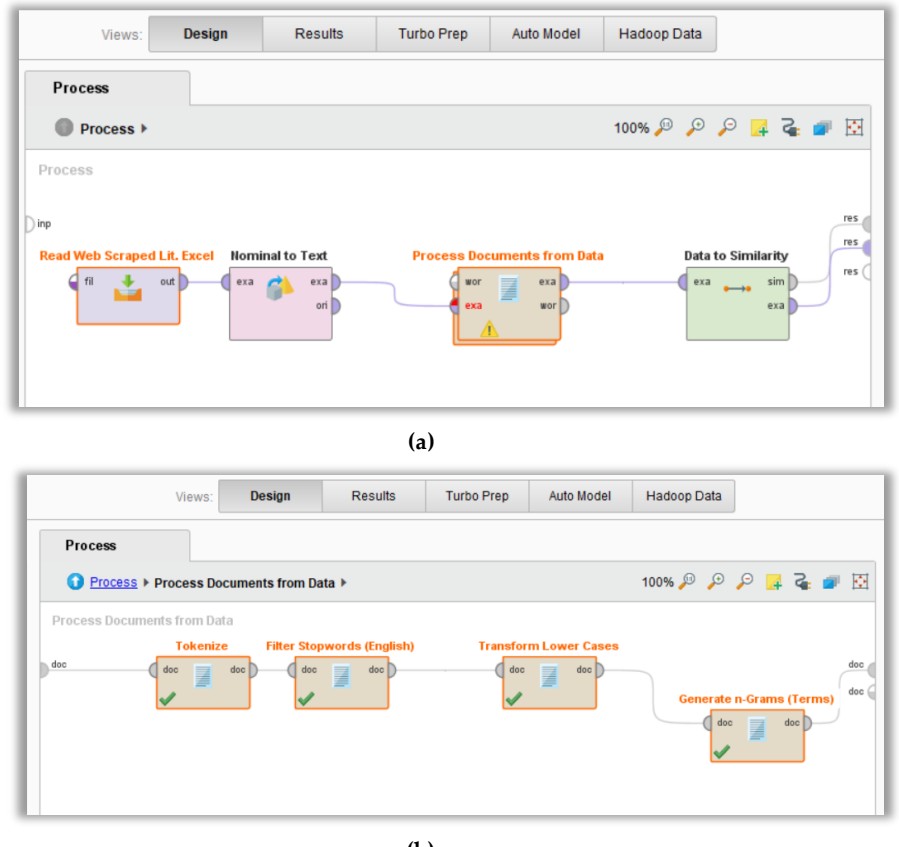

**Figure 4.** (**a**) Rapidminer process model deriving term similarity from the scraped literature search; (**b**) text-mining sub-process components (tokenize, filter stop words, transform lower case and generate n-grams).

It could be observed that all of the below components are GUI-based components that work on increasing the ease of use as well as time to value from the tool.

## 5. Case Study

The following case study helps us substantiate the proposed methodology and methods with a focus on Steps 2–4 and how these steps affect further steps within the methodology (Steps 5 and 6), with an aim towards augmenting systematic reviews through the relevant literature search on specific topics. Correspondingly, a play role approach is adopted to conduct the case study [20]. The context here is set by addressing how a researcher would be able to conduct a systematic review by probing a specific topic (in this case: "enterprise systems challenges"); for which keywords would be used as the search criteria across the eight basket journals of the *Association of Information Systems* (please see Appendix A.4) on a repository like Google Scholar. It is to be noted that the main purpose of this case study is to inquire how earlier explained methods' ease of use would contribute to mass adoption in similar scholar investigations and systematic reviews.

It begins with Step 1 (inquiring a research concept based on understanding), where the trigger towards inquiring the journal basket through the researcher's prior knowledge of the following facts relating to "enterprise systems challenges". It could be observed from the below facts that a high emphasis on failures taking place during the ERP implementation and post-implementation phases:

*ERP systems have been criticized for not maintaining the Return-on-Investments (RoI) promised. Sykes et al. claims that 80% of ERP implementations fails [21]. Also, 90% of large companies implementing ERP-systems failed in their first trial [22]. It has also been reported that between 50%–75% of US firms experience some degree of failure. Additionally, 65% of executives believe ERP implementation has at least a moderate chance of hurting business [23].*

*Three quarters of ERP projects are considered failures and many ERP Projects ended catastrophically [24]. Failure rates estimated to be as high as 50% of all ERP implementations [25]. And as much as 70% of ERP implementations fail to deliver anticipated benefits [26]. Still many ERP systems still face resistance and ultimately failure [27,28]. That been said, there are two main critical phases that might lead to the failure of an ERP project. The Implementation as well as the post-implementation phases marks to two main phases in a given ERP lifecycle where many of the organizations might experience failure. The two phases include similar activities and involve similar stakeholders.*

This would then lead to second step where the researcher probes the initial discovery points in order to conduct the systematic review to either back up or refute his prior understanding around challenges being faced with enterprise systems. The requirement for this step is to query the scientific repository (i.e., Google Scholar in this case) to initially make sense of the outcomes, and then to make sure that whatever is being returned falls within our research domain. The search criteria and keywords used in this case were: {"enterprise systems" challenges}. If the search hits for the corresponding keywords being used are not efficient, then rather more significant keywords would be used.

Once the search hits are validated, the third step of scraping and extracting metadata points (like literature titles, publication year, etc.) from the search body would be built and executed to produce. The earlier web scraping method would be employed to gather metadata of interest from all of the eight basket outlets by building web-scrapers using the graphical UI to produce the hits in Table 1. Figure 5 on the other hand shows the output from this exercise is extracted in csv format serving as the input to the extractive summarization for relevant literature in the next step. Based on this method a considerable number of titles is extracted to be used for further analysis.

**Table 1.** List of Association of Information Systems (AIS) basket journals and corresponding search hits from Google Scholar for enterprise systems challenges.

| No | Outlet/Journal Name | Search Hit Count |
|----|---------------------|------------------|
| 1 | Journal of the Association of Information Systems | 34 |
| 2 | Information Systems Journal | 21 |
| 3 | Information Systems Research | 40 |
| 4 | Journal of Information Technology | 111 |
| 5 | Journal of Management Information Systems | 26 |
| 6 | Journal of Strategic Information Systems | 23 |
| 7 | Management Information Systems Quarterly | 61 |
| 8 | European Journal of Information Systems | 45 |
| | **Total** | **361** |

| web-scrap | web-scrap | title | | page | page-href |
|-----------|-----------|-------|-|------|-----------|
| 1554519 | https://sch | Proposing the multi-motive information systems continuance model (MISC) to better explain end-user sy | Next | https://scholar-google-se.proxy.lib.ltu.se/ |
| 1554519 | https://sch | Rejoinder to the Response to" The Scholarly Capital Model" | | | |
| 1554519 | https://sch | Helpfulness of Online Review Content: The Moderating Effects of Temporal and Social Cues | Next | https://scholar-google-se.proxy.lib.ltu.se/ |
| 1554519 | https://sch | The impact of functional affordances and symbolic expressions on the formation of beliefs | Next | https://scholar-google-se.proxy.lib.ltu.se/ |
| 1554519 | https://sch | Use of online social networking services from a theoretical perspective of the motivation-participation-p | Next | https://scholar-google-se.proxy.lib.ltu.se/ |
| 1554519 | https://sch | A theory of organization-EHR affordance actualization | Next | https://scholar-google-se.proxy.lib.ltu.se/ |
| 1554519 | https://sch | The role of business intelligence and communication technologies in organizational agility: a configuratio | Next | https://scholar-google-se.proxy.lib.ltu.se/ |

**Figure 5.** Sample CSV output for extracting paper titles from Google Scholar.

The process of modeling the extractive summarization for relevant literature for Step 4 of the methodology comes in to execute a set of text mining algorithms as illustrated in Figure 6a,b where the term-based method is employed for information extraction and clustering [29]. Despite the polysemy

and synonymy issues with the term-based methods, this approach is adopted given the nature of literature search activities of using keywords (as "terms") to inquire an academic concept. The choice of which algorithms to use is quite dependent on the understanding of how they work. For instance, cosine similarity is best suited for text similarity and clustering which is the measure used in our case [30].

The researcher executes X-means clustering and random forest algorithms to help further translating text population and corresponding relations [31]. These collective algorithms help significantly in the next steps of the adopted methodology, where further evaluation of the most relevant literature is conducted as part of Steps 5 and 6. The nature of the methodology adopts an iterative approach to ensure refinement of the research understanding that leads to employment-extended research activities effectively.

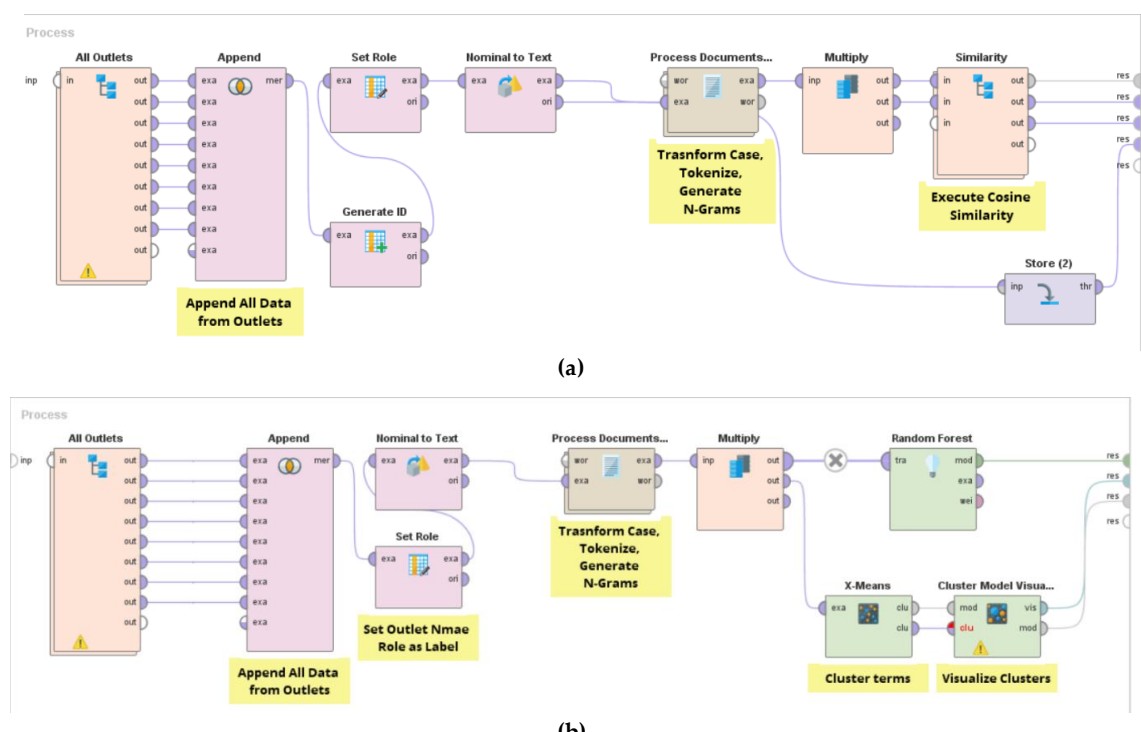

**Figure 6.** (**a**) Process model for Information Extraction and Similarity on all literature titles; (**b**) Process model for X-Means clustering and Random Forest.

*Results*

Interpretation and evaluation of the returned results from the earlier exercise is highly dependent upon the researcher's analysis and synthesis of the literature as well as academic domain understanding, which is linked to Steps 5 and 6 of the methodology. As explained, the iterative nature of systematic reviews and the methodology at hand puts research rigor at a high rank for an academic concept to be either proven or refuted. The results at which the researcher had discovered helps significantly support these decisions. In this case it had been initially discovered by the researcher that leading causes behind enterprise systems challenges are implementation and post-implementation bottlenecks. This required further investigation. The below results show the heatmap produced out of the X-means clustering algorithm (Figure 7).

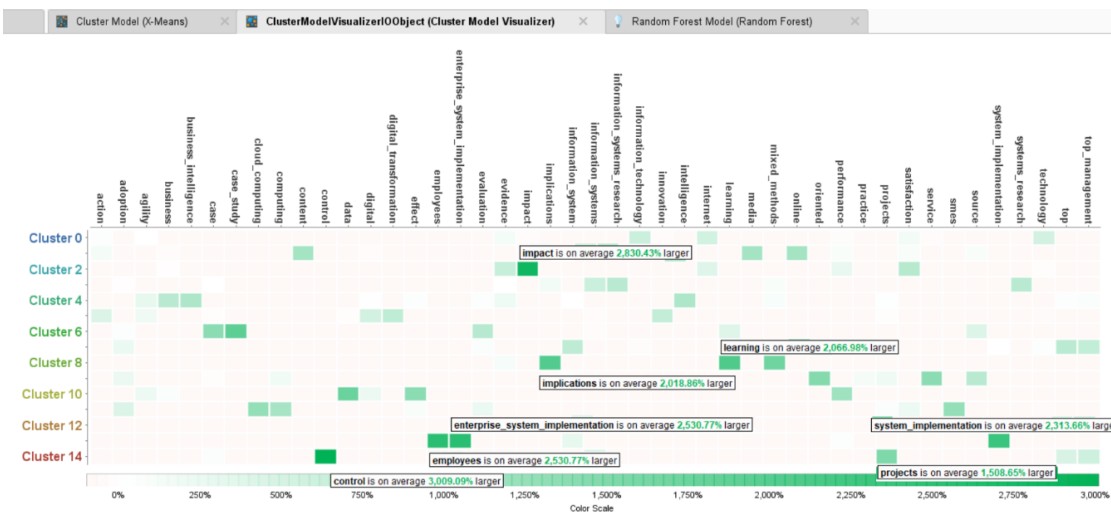

**Figure 7.** Visualizing X-means algorithm using a heatmap graph for the top terms representing the clusters.

It could immediately be observed that for the top analyzed terms being analyzed, they are clustered around terms like: {*control, impact, employees, enterprise_system_implementation, system_implementation, learning, implications, projects, etc.*}. It however is not the case for post-implementation. By examining several decision trees produced by the random forest algorithm, it could be observed from the post-implementation, despite making a small fraction of appearances across the literature scraped from the journals (Figure 8), that it still surfaces across many of these trees (Figure 9).

| Row No. | word | in documents | total | in class (MIS Quarterly) | in class (Journal of IT) | in class (European Journal of IS) | in class (Jo... | in |
|---------|------|--------------|-------|--------------------------|--------------------------|-----------------------------------|-----------------|-----|
| 199 | perspective | 24 | 24 | 1 | 9 | 3 | 3 | 1 |
| 200 | phase | 3 | 3 | 0 | 1 | 1 | 0 | 0 |
| 201 | planning | 6 | 6 | 1 | 5 | 0 | 0 | 0 |
| 202 | planning_erp | 4 | 4 | 0 | 4 | 0 | 0 | 0 |
| 203 | platform | 4 | 4 | 0 | 1 | 1 | 0 | 1 |
| 204 | post | 7 | 7 | 2 | 2 | 1 | 1 | 0 |
| 205 | post_implementation | 5 | 5 | 2 | 2 | 1 | 0 | 0 |
| 206 | potential | 5 | 5 | 1 | 1 | 0 | 1 | 0 |
| 207 | power | 3 | 3 | 1 | 0 | 1 | 0 | 1 |
| 208 | practice | 7 | 7 | 0 | 3 | 2 | 0 | 0 |
| 209 | practices | 7 | 7 | 0 | 2 | 0 | 0 | 1 |

**Figure 8.** Post-Implementation term is ranked the 205th most frequent term from the generated N-Grams, with only five appearances in total.

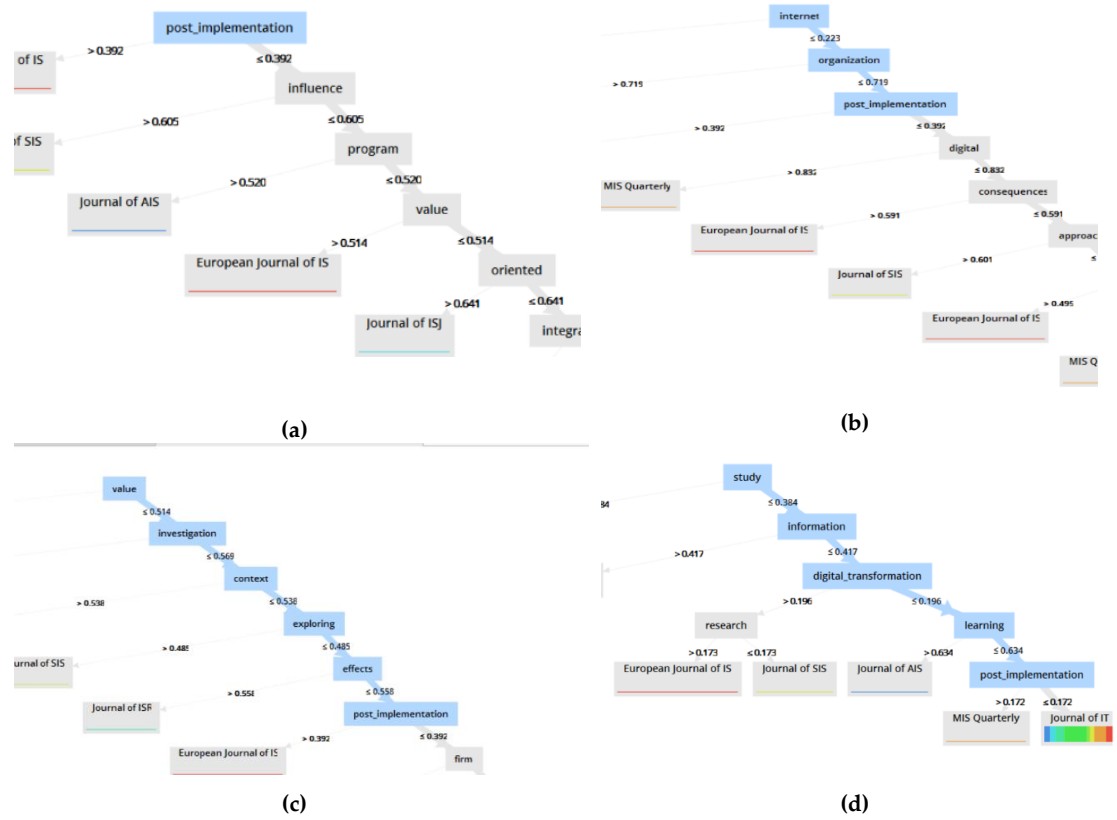

**Figure 9.** "Post-Implementation" term appearing in several trees produced by the Random Forest algorithm.

By implementing the above methods; results show proof towards less literature counts on "post-implementation" lifecycles. Despite the fact, this phase within the ERP lifecycle typically poses significant bottlenecks that could impede digital transformation initiatives as illustrated in Figure 9d. The researcher then has evidence that the post-implementation phase has high affinity to prove the initial hypothesis. Yet there might not be much literature focus on the topic, which could direct his research attention towards focusing upon that topic. Through cosine similarity (Figure 10) the researcher is also able to link the highly relevant literature to the topic of interest for which enterprise systems challenges could be rigorously researched and tackled.

**Figure 10.** Relevant Post Implementation literature titles with cosine similarity measures.

Correspondingly, the results did show how the proposed methodology could present a stance towards augmenting the activities of a relevant literature search. Hence, enabling researchers to easily and efficiently achieve the objectives of breadth and depth of knowledge, rigor and consistency of comprehension and application, as well as the clarity and brevity of analysis, synthesis and evaluation.

To further assess the performance of the artifact at hand; the author of this scholar work conducted a literature review as part of another work in progress research activity examining the same topic of "enterprise systems challenges". Details of this literature review exercise are mentioned in Appendix B.

Out of the literature review table illustrated in Appendix B, the total count, as well as percentages of the total included articles, are calculated for every phase and dimension within the Enterprise System (ES) implementation lifecycle. Table 2 correspondingly shows high affinities towards Implementation as well as Use and Maintenance phases (with 49% and 66%, respectively, from the manually shortlisted content of the literature review exercise). On the other hand, People and Process dimensions (86% and 46%, respectively, from the manually shortlisted content of the literature review exercise) are ones that convey review focus.

**Table 2.** The total count as well as percentages of total included articles for every phase and dimension within the ES implementation lifecycle.

| | Adoption | Acquisition | Implementation | Use & Maintenance | Evolution | Retirement | Change Management | People | Process | Product |
|---|---|---|---|---|---|---|---|---|---|---|
| **Total Count** | 3 | 0 | 17 | 23 | 1 | 1 | 5 | 30 | 16 | 2 |
| **% of Total** (out of 34 shortlisted articles based on content) | 9% | 0% | 49% | 66% | 3% | 3% | 14% | 86% | 46% | 6% |

By examining Figure 7 through Figure 10 versus the table produced by the typical literature review exercise, it could be observed that the use of the CRISP-DM methodology at hand does set the compass towards the right direction. Congruently, the observed key term clusters, as well as the random forest trees, suggest a focus towards a potential topic conceptualization combining the top observed key terms as follows:

- The "impact" and "implications" of "enterprise_system_implementation" "projects" and "post_implementation" phases on organizational "control", "employees" and "learning".

## 6. Scientometric Relatedness

The task of defining the review scope (as part of von Brocke's literature review process) and predicting the expected impact from the to-be-conducted review activity, researchers typically inspect the scientific contribution of authors, journals or specific works, as well as the analysis of the dissemination process of scientific knowledge. Archetypal indices such as *h*-index [32], *g*-index [33] and many others are used for these types of assessment. These indices typically model knowledge exchanges in scientific discourses, which correspondingly demonstrates a high relatedness to the topic at hand. However, knowledge exchanges in scientific discourses cannot be reduced to the exchanges of information in co-authorship, co-word, or citation relations [34]. This very fact explains how the methodology at hand rather complements the various scientometric indices with a focus on uncovering co-word and citation relations serving to augment the literature review process. This puts a granular focus on revealing latent structures within the to-be-cited content and journal texts. Correspondingly, it could be argued that the methodology at hand tends to focus on macro relations, whereas larger scale frameworks like Harzing's Publish or Perish framework tends to focus on micro scale relations [35].

The following list of norms compare some of the top traits comparing the methodology at hand with the example of Harzing's Publish or Perish (HPoP) Framework:

- Framework Focus:

    ○ *Scale focus of the framework; Micro or Macro.*

- Ease of Use:

    ○ *User friendliness*

- Flexibility

  - *Enabling cross-functional, multidimensional and multi-algorithmic analysis*

- Reproducibility

  - *Ability for users to obtain the same results using authors' own analyzed data*

- Replicability

  - *Ability for users to obtain substantially similar results by applying the same steps in a different context with different data*

- Reach

  - *Ability for users to include other data sources*

- Data Preparation Need

  - *Requirement for users to thoroughly prepare datasets to conduct the analysis exercise*

- Visual Capability

  - *Users ability to visualize the analyzed citation data*

That being said, Table 3 illustrates the main differences based on the above criteria between the methodology at hand and Harzing's Publish or Perish.

**Table 3.** Comparing CRISP-DM approach Harzings' Publish or Perish.

| Criteria/Methodology | CRISP-DM | Harzing's Publish or Perish (HPoP) |
|---|---|---|
| **Framework Focus** | Macro scale<br>*Conducts the analysis exercise on word level relations of the body of science* | Micro scale<br>*Conducts the analysis over scientific discourse in relation to authors, journals and scientific citations* |
| **Ease of Use** | High<br>*Depends on the user data science skills and understanding* | High Ease of Use<br>*Educates users on the software use through user guides* |
| **Flexibility** | High<br>*Data Mining software provides a plethora of algorithms to be conducted over multi-dimensional and cross functional data* | Medium<br>*Bound by the features of the HPoP software* |
| **Reproducibility** | High<br>*Models can be exported and shared with users easily* | High<br>*Steps to reproduce the analysis can be easily followed using the software GUI* |
| **Replicability** | Medium<br>*Ability to replicate models depends on users' skills* | High<br>*Steps to replicate the analysis can be easily followed using the software GUI* |
| **Reach** | High<br>*Provides unlimited reach for users to further expand on the analysis of their data* | Low<br>*Provides limited analysis channels (Google Scholar, Web of Science, Scopus, etc.)* |
| **Data Preparation Need** | Low<br>*Requires the use of web scraping methods to extract and prepare data* | High<br>*Does not posture any data preparation needs by the users* |
| **Visual Capabilities** | High<br>*Data Science tools typically have a plethora of visualization tools enabling users to model understandable assessment visualizations* | Low<br>*Requires use of additional components to produce visually meaningful diagrams (e.g., VOSViewer, CiteNetExplorer, etc.)* |

## 7. Future Work and Limitations

It could be then deducted that data mining tools and methods used to instigate the CRISP-DM methodology complement the HPoPs' ability to augment and set research compasses, specifically when it comes to conducting a systematic literature review. However, there remain a set of aspects that needs to be addressed before the augmentation bottlenecks are dsicussed.

The first aspect would be the requirement to repackage a product that similarly works like the HPoP software. It could be observed that both methods employ very similar underlying tools (data preparation/scraping, statistical components, etc.) in order to reach out to the derivation of both micro and macro level analysis of the citations of interest. Correspondingly, a packaged end-to-end CRISP-DM framework would be significantly important to provide the sought-after outcomes without complications that might arise from the drawbacks mentioned within Table 3.

Owing to the fact that the methodology at hand seeks to push the boundaries of research activities like literature reviews, the need to reserve the priority to furnish superior data formats demands an undertaking. This should be prioritized before further delving into research discussions on how more sophisticated algorithms could be used to resolve the manual efforts accompanied with augmentation activities. For that matter, domain ontologies would be the key to catalyze the task of automatic knowledge extraction [36,37]. The need for domain ontologies furnishes the foundation for the solution packaging between the tools to read and understand aspects of interrelation, instantiation, subsumption, exclusion, axiomization and attribution [38].

## 8. Conclusions

In this paper we did show how a literature search could be augmented through applying CRISP-DM as well as overcoming the typical data extraction issues from scientific web repositories using simplified web scraping methods. This then caters for a rather augmented process of relevant literature search and summarization by researchers within a literature review context. The GUI-based interfaces of the adopted methods could furnish the base for a mass adoption of the proposed methodology, which would contribute significantly towards simplifying typical mundane literature review objectives in a more scalable and efficient manner.

The data extraction issue had been addressed in this paper by using "Web Scraper", which is a GUI-based extension to Google Chrome to scrape information out of the literature search process. That significantly saves a considerable amount of time furnishing this data to be passed onto the augmented relevant literature search and summarization process. This solution then extends the adoption of the methodology by extending the reach to scrape information from the wider base of scientific repositories that do not necessarily follow open access standards. Correspondingly, this augments the complex process of our literature search. On the other hand, this opens the discussion of how scientific web repositories can play a major role into catalyzing the adoption of the proposed CRISP-DM methodology for relevant literature search. Correspondingly, this paper urges these repositories to furnish channels at which citation metadata (titles, abstracts, outlet names, etc.) could be extracted easily. Hence, contributing to the reproducibility and replicability of various research activities dependent on the analysis of such information.

The paper had also illustrated the use of Rapidminer as a data mining tool; which is an easy to use yet robust advanced analytics platform that works on applying modeling steps of CRISP-DM through executing the cosine similarity algorithm as an example. The aim was to explain to the wider research community how similar tools could be easily used to significantly increase productivity and maintain a level of flexibility from an analysis perspective of the relevant searched literature.

The conducted case study did work on applying the proposed methodology and methods by adopting a play role approach, studying observations of a researcher with an interest in tackling challenges-related enterprise systems. The researcher had an initial hypothesis that required proof to continue pursuing the topic with an emphasis on studying the post-implementation phase. By applying text similarity, clustering and the random forest algorithm, the results did show that there was not

much scholar work conducted focusing on the topic. However, the topic is pretty much related to several key concepts that trigger the requirement for further research rigor through analyzing the relevance with other literature. Consequently, the case did show how augmenting a relevant literature search using CRISP-DM could be an effective and efficient approach towards handling and tackling systematic reviews.

On the other hand, the author compared the output from the data mining tool with a manual literature review exercise conducted as part of a work in progress research (at the time this paper was published). This led congruent outcomes that showed overlapping outcomes, setting research compasses towards the same directions.

This research activity also concludes by indicating two main future research works required to be conducted in order to complete this complementary approach of from an automatic knowledge extraction. First of which is the need to further repackage the conducted methods within this research in a GUI-based product that enables unskilled researchers to reuse the elucidated methods. The second on the other hand discusses the need to employ concepts of domain ontologies that enhances the level at which information is to be used and evaluated by humans and machines to extract relevant representations over topic categories, properties and relations. Which correspondingly seeks towards pushing the augmentation abilities of the CRISP-DM methodology at hand for extractive summarization.

**Funding:** This research received no external funding.

**Acknowledgments:** The author would like to acknowledge Rapidminer for providing the necessary Rapidminer educational licenses. These licenses had been leveraged to build the research artifact contributing to the evaluation of our scholar work.

**Conflicts of Interest:** The author declares no conflict of interest.

## Appendix A

*Appendix A.1 Semantic Scholar Open Research & Corpus Download Examples:*

Semantic scholar is a corpus database that is restricted to open access research publications. The platform exposes both data samples, or the full data set through either REST API command (wget) or using Amazon Web Services—Command Line Interface (AWS-CLI).

- REST API: wget –I, Available online: https://s3-us-west-2.amazonaws.com/ai2-s2-research-public/open-corpus/manifest.txt (accessed on 3 December 2019);
- AWS CLI: aws s3 cp –recursive, Available online: s3://ai2-s2-research-public/open-corpus/destinationPath (accessed on 3 December 2019).

*Appendix A.2 xPath:*

- XPath is defined as XML path. It is a syntax or language for finding any element on the web page using XML path expression. XPath is used to find the location of any element on a webpage using HTML Document Object Model (DOM) structure

*Appendix A.3 Web Scraper:*

- Web Scraper Web Site, Available online: https://www.webscraper.io/ (accessed on 3 December 2019);
- Google Chrome Extension: Web Scraper, Available online: https://goo.gl/QuUdHu (accessed on 3 December 2019).

*Appendix A.4 Association of Information Systems Journals Basket:*

- European Journal of Information Systems
- Information Systems Journal
- Information Systems Research

- Journal of AIS
- Journal of Information Technology
- Journal of MIS
- Journal of Strategic Information Systems
- MIS Quarterly

*Appendix A.5 Git Repository Location- CRISP-DM for Augmented Literature Search:*

Available online: https://github.com/hhassanien/CRISP-DM-for-Augmented-Relevant-Literature-Search (accessed on 3 December 2019).

**Appendix B**

This exercise conducts a typical literature review as part of a Work in Progress research activity that adopts von Brocke's [1] methodological framework, as illustrated in Figure A1 with the corresponding steps: (1) definition of review scope (2) conceptualization of the topic (3) literature search (4) literature analysis and synthesis (5) research agenda. The following subsections explain the steps taken in the literature review process following these steps.

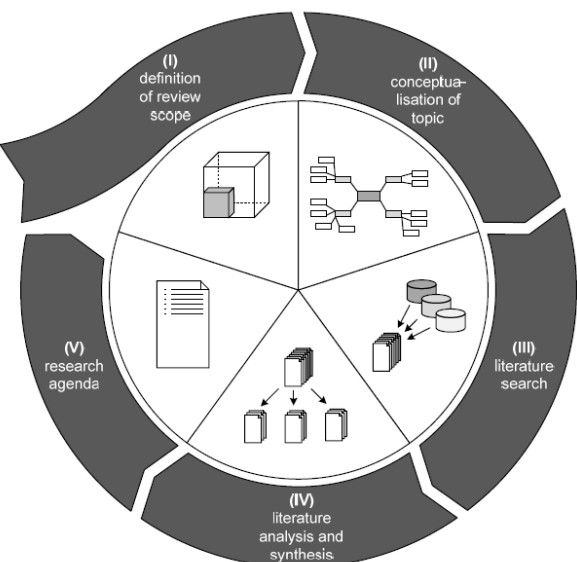

**Figure A1.** Illustration of von Brocke's Literature Review Framework [1].

With a focus on the Esteves and Pastor [39] model as depicted in Figure A2, the implementation lifecycle stages and dimensions:

- Stages:

  ○　Adoption Decision
  ○　Acquisition
  ○　Implementation
  ○　Use and Maintenance
  ○　Evolution
  ○　 Retirement

- Dimensions:

  ○　People
  ○　Process

- ○ Product
- ○ Change Management

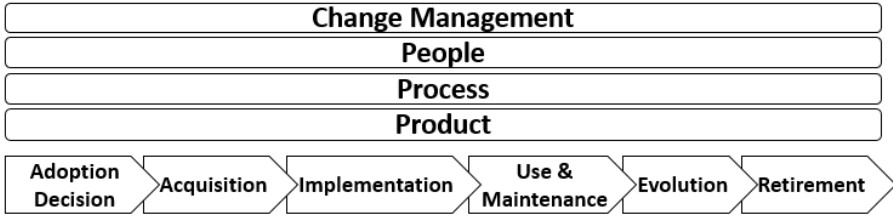

**Figure A2.** The ES Implementation Lifecycle [39].

As explained in earlier sections, the scope at hand reviews challenges in enterprise systems across the ES implementation lifecycle stages and dimensions, for which the following relevant reviewed literature are illustrated in Table A1:

**Table A1.** The expanded view of Table 2 referred to within the text. This shows the included/cited journal articles and the corresponding topic focus across the ES implementation lifecycle phases (highlighted in blue) and dimensions (highlighted in orange) marked in XX.

| # | Outlet | Reference | Adoption | Acquisition | Implementation | Use & Maintenance | Evolution | Retirement | Change Management | People | Process | Product |
|---|--------|-----------|----------|-------------|----------------|-------------------|-----------|------------|------------------|--------|---------|---------|
| 1 | JAIS | [40] | | | XX | XX | | | | XX | XX | |
| 2 | JAIS | [41] | | | | XX | | | | XX | | XX |
| 3 | JAIS | [42] | | | XX | XX | | | | XX | XX | |
| 4 | JAIS | [43] | | | XX | XX | | | | XX | | XX |
| 5 | JAIS | [44] | | | XX | | | | XX | XX | | |
| 6 | JAIS | [45] | | | | | XX | | | XX | | |
| 7 | ISJ | [46] | XX | | | XX | | | | XX | | |
| 8 | ISJ | [47] | | | | | | | | XX | | |
| 9 | ISJ | [48] | | | XX | XX | | | | | XX | |
| 10 | ISJ | [49] | | | XX | XX | | | | XX | | |
| 11 | ISJ | [50] | | | | XX | | | | XX | XX | |
| 12 | ISR | [51] | | | | XX | | | | XX | | |
| 13 | ISR | [52] | | | | XX | | | | XX | XX | |
| 14 | ISR | [53] | | | | XX | | | | XX | XX | |
| 15 | ISR | [54] | | | XX | XX | | | | XX | | |
| 16 | JIT | [55] | | | XX | XX | | | | | XX | |
| 17 | JIT | [56] | | | XX | | | | XX | XX | | |
| 18 | JIT | [57] | | | | | XX | | | XX | | |
| 19 | JIT | [58] | | | XX | | | | XX | XX | | |
| 20 | JIT | [59] | | | | XX | | | | XX | | |
| 21 | JIT | [60] | | | | XX | | | XX | XX | XX | |
| 22 | JMIS | [61] | | | | XX | | | | XX | XX | |
| 23 | JMIS | [62] | XX | | | XX | | | | XX | | |
| 24 | JSIS | [63] | | | XX | XX | | | | XX | XX | |
| 25 | JSIS | [64] | | | XX | XX | | | | XX | | |
| 26 | JSIS | [65] | | | XX | | | | | | XX | |
| 27 | MISQ | [66] | | | | XX | | | | XX | XX | |
| 28 | MISQ | [21] | | | | XX | | | | XX | XX | |
| 29 | MISQ | [67] | | | XX | | | | | XX | | |
| 30 | MISQ | [68] | | | XX | | | | | XX | XX | |
| 31 | MISQ | [69] | XX | | XX | | | | | XX | | |
| 32 | MISQ | [70] | | | | XX | | | | XX | XX | |
| 33 | EJIS | [71] | | | XX | | | | XX | XX | | |
| 34 | EJIS | [72] | | | | XX | | | | XX | | |

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
