# Peer review of "Web Scraping Scientific Repositories for Augmented Relevant Literature Search Using CRISP-DM"

_asi, doi:10.3390/asi2040037_

Round 1
Reviewer 1 Report
The figures should be more clear, I think following the paper is hard (a pipeline could be added)
Author Response
Dear Reviewer,
I would like to thank you very much for the valuable comments being shared in correspondence to our research publication now being titled as “Web Scraping Academic Repositories for Augmented Relevant Literature Search using CRISP-DM”. It is without a doubt that the comments being provided had contributed to enhancing the quality of our academic work on a number of fronts.
These constructive suggestions for enhancements had contributed towards increasing the quality of a number of key pointers. We have responded to the steps taken in order to make sure the comments were addressed which could be further reviewed and assessed as per the attached spreadsheet.
That being said, we’d like to thank you all again for giving us the chance to re-submit our publication hoping that we’d hear from you again the positive news.
Please find the below responses to the comments being provided:

Reviewer 2 Report
The article proposes using web scraping as well as advanced analytics methods for relevant literature search within the research community.
Comments:
1. The novelty of the article is not clear: the topic was already addresses in ref. [10].
2. The paper does not discuss the systematic review methods such as proposed by B Kitchenham, and the review reporting guidelines such as Preferred Reporting Items for Systematic Reviews and Meta-Analyses (PRISMA). The graph analysis and visualization tools such as CiteSpace and VOSViewer tools also are not mentioned.
3. Description of research methodology is not clear: provide a block diagram or a flow diagram of your method.
4. Figure 1: check quality, resolution is too low.
5. The results part of the paper are missing: I suggest adding a large case study to compare with the search results provided, e.g., by Google Scholar.
6. The limitations of the method should be discussed. A comparison with the numerical results in information retrieval achieved by other methods should be provided.
7. Conclusions are too vague.
8. Check quality of references: many have bibliographic information (journal names, pages, etc.) missing.
9. Most of cited references are outdated. Check the recent publications on the advances in the field of literature review such as: Uddin, S., Choudhury, N. & Hossain, M.E. Scientometrics (2019). https://doi.org/10.1007/s11192-019-03057-4
Author Response

(The authors gave the same response as above.)

Round 2
Reviewer 2 Report
1. Motivation and the need of web scraping for literature search is not clear. The authors should have presented a case study demonstrating that scraping allows to extract more information than using established bibliographic databases such as Web of Science or Scopus.
2. The paper lacks of methodological background. The authors mention guidelines in [1], however it is not clear how the process model proposed in this paper integrates or goes beyond the already established literature review guidelines.
3. Diagram in Figure 1 is not formal. The meaning of different types on entities and arrows is not explained. “Data” block stands alone, so no data is used and/or produced by the process.
4. The main drawback of the paper is that a large-scale case study that would validate the proposed process model is missing.
5. Conclusions are a set of assumptions that the proposed model or method “could” or “would” do. Without a large-scale case study, there is no guarantee that the described method or model works and is useful.
6. No significant updates to reference list (only a few items changed). There are many relevant articles published by Publications and Scientometrics journals (as well as others) that fit the topic of this article.
Author Response
1. Motivation and the need of web scraping for literature search is not clear. The authors should have presented a case study demonstrating that scraping allows to extract more information than using established bibliographic databases such as Web of Science or Scopus.
A Case had been presented in order to demonstrate the motivation and the need.
2. The paper lacks of methodological background. The authors mention guidelines in [1], however it is not clear how the process model proposed in this paper integrates or goes beyond the already established literature review guidelines.
CRISP-DM had been adopted as a methodology. Updated figure for the methodology had been added
3. Diagram in Figure 1 is not formal. The meaning of different types on entities and arrows is not explained. “Data” block stands alone, so no data is used and/or produced by the process.
Updated figure for the methodology had been added
4. The main drawback of the paper is that a large-scale case study that would validate the proposed process model is missing.
A large scale case study had been conducted in order to validate the proposed model
5. Conclusions are a set of assumptions that the proposed model or method “could” or “would” do. Without a large-scale case study, there is no guarantee that the described method or model works and is useful.
A large scale case study had been conducted in order to back up the assumptions with facts
6. No significant updates to reference list (only a few items changed). There are many relevant articles published by Publications and Scientometrics journals (as well as others) that fit the topic of this article.
References had been amended in order to fit the case study

Round 3
Reviewer 2 Report
The authors should discuss automatic knowledge extraction from literature search results such as domain ontologies, which allow both comprehensive description of domain concepts and their respective relations as well as for visualization. Use the following sources:
Automatic generation of concept taxonomies from WEB search data using support vector machine. Proceedings of the 5th International Conference on Web Information Systems and Technologies, 673-680.
Automatic generation of part-whole hierarchies for domain ontologies using web search data. 32nd International Convention Proceedings: Computers in Technical Systems and Intelligent Systems, 3, 215-220.
‘Research method’ section must be improved. Provide a flow diagram (or algorithm) of your method.
The results must be compared with the ones obtained other similar tools or methods such as Harzing’s Publish or Perish. To evaluate and compare the results, use information retrieval metrics such as precision, recall, Precision at k documents (P@k), R-precision, Discounted cumulative gain, etc.
RapidMiner should be compared with other alternatives (including Harzing’s Publish or Perish) according to a set of predefined criteria.
Author Response
Dear Editors and Reviewers,
First off, I'd like to present my sincere apologies with regards to the delayed resubmission that corresponds to the latest review inputs. However, throughout the intermediary period of time, both research and force majeure related circumstances had been encountered that contributed to the delay.
In correspondence to the research related circumstances, it had been identified that further parallel research activities had deemed to be required in order to shed more light onto this being reviewed scholar work (please see Appendix B). That being said, please find the responses to the very valuable comments being provided by the reviewer which significantly contributed towards the enhancement of this research work.
The authors should discuss automatic knowledge extraction from literature search results such as domain ontologies, which allow both a comprehensive description of domain concepts and their respective relations as well as for visualization. Use the following sources: Automatic generation of concept taxonomies from WEB search data using support vector machine. Proceedings of the 5th International Conference on Web Information Systems and Technologies, 673-680. Automatic generation of part-whole hierarchies for domain ontologies using web search data. 32nd International Convention Proceedings: Computers in Technical Systems and Intelligent Systems, 3, 215-220.
Please see the Future Work and Limitations Section
‘Research method’ section must be improved. Provide a flow diagram (or algorithm) of your method.
Please see figure 1
The results must be compared with the ones obtained other similar tools or methods such as Harzing’s Publish or Perish. To evaluate and compare the results, use information retrieval metrics such as precision, recall, Precision at k documents (P@k), R-precision, Discounted cumulative gain, etc.
Please see the Future Work and Limitations Section
RapidMiner should be compared with other alternatives (including Harzing’s Publish or Perish) according to a set of predefined criteria.
Please see the scientometrics relatedness Section
Additionally, the parallel conducted research work manifested in a work in progress literature review exercise show how congruent outcomes show overlapping outcomes setting research compasses towards the same directions between the augmented extractive summarization approach versus a manual approach conducted within the parallel work in progress work.
That all being said, I’d like to thank you very much again for valuable inputs and patience.
Regards,
The Author
